# Ginseng-Based Nanotherapeutics in Cancer Treatment: State-of-the-Art Progress, Tackling Gaps, and Translational Achievements

**DOI:** 10.3390/cimb47040250

**Published:** 2025-04-03

**Authors:** Pragya Tiwari, Kyeung-Il Park

**Affiliations:** Department of Horticulture and Life Science, Yeungnam University, Gyeongsan 38541, Republic of Korea; pki0217@yu.ac.kr

**Keywords:** drug delivery system, ginseng-based carbon dots (GCDs), nanotherapeutics, *Panax ginseng*, sustainable development goals (SDGs), triterpenoids

## Abstract

Among medicinal plants, the *Panax* genus (family: *Araliaceae*) includes plant species widely recognized for their multi-faceted pharmacological attributes. The triterpenoids, designated as ginsenosides, are increasingly recognized as drug-like molecules in cancer therapies due to their therapeutic role in restricting tumor invasion, proliferation, metastasis, apoptosis, and drug resistance reversal in tumor cells. In the nanobiotechnological era, nano-delivery systems provide feasible solutions to address bottlenecks associated with traditional drug delivery methods (low bioavailability, instability in the gastrointestinal tract, high dosage requirements, side effects, poor absorption, and incomplete drug utilization in the body). The dedicated efforts for precise and effective treatment have directed the development of ginsenoside-based nano-delivery systems to achieve potent anticancer efficacies and address the limitations in ginseng pharmacokinetics, facilitating drug development trials. Studies into ginseng pharmacokinetics showed a remarkable prolonged clearance and free drug levels of ~15% (ginsenoside RB1 nanoparticles) in mice (compared to only ~5% for ginsenosides) and better antitumor efficacies, demonstrating key success in ginseng biotechnology for drug development. Delving into the nanobiotechnological interventions in ginseng-derived therapeutics, this study summarizes current advances and achievements, particularly in cancer treatment, tackles existing gaps, focuses on feasible solutions, and examines prospects of translational success.

## 1. Introduction

In the era of unprecedented climate change and rising global population, the Sustainable Development Goals (SDGs) advocate harnessing bio-based resources for addressing malnutrition and poverty, improving environmental and human well-being, and driving economic progress [1,2]. Herbal medicines are fast gaining recognition as potential drugs with good efficacies and less side effects compared to conventional medicines. The *Panax* genus has a widespread and rich history of use in the traditional system of medicine and is classified in the family *Araliaceae* [3]. To date, 17 *Panax* species including *Panax ginseng*, *Panax notoginseng*, *Panax trifolius*, *Panax sikkimensis*, *Panax quinquefolius*, *Panax bipinnotifidus*, *Panax major*, *Panax quineuefolius*, *Panax vietnamensis*, *Panax zingiberenensis*, *Panax bipinnotifidus*, *Panax elegantior*, *Panax omeiensis*, *Panax stipuleanatus*, *Panax sinensis*, *Panax japonicus*, *Panax pseudoginseng*, *Panax zingiberenensis*, and *Panax wangianus* are reported in the literature and grown in different geographical regions worldwide [4,5]. The plant attributes its pharmacological properties to the presence of unique triterpenoids, designated as ginsenosides. The bioactive components, ginsenosides, are classified as oleanane and dammarane saponins according to their aglycone moiety [6]. The dammarane skeleton is present in most ginsenosides and comprises 17 carbons in a four-ring structure with various sugar moieties (e.g., glucose, rhamnose, xylose, and arabinose) attached to the C-3 and C-20 positions [7]. Among different types, ginsenosides are either classified as (a) 20(S)-protopanaxatriol (PPT) (Re, Rf, Rg1, Rg2, Rh1) or (b) 20(S)-protopanaxadiol (PPD) (Rb1, Rb2, Rb3, Rc, Rd, Rg3, Rh2, Rs1) [8]. Till now, approx. 200 ginsenosides have been documented, with the major ginsenosides including Rb1, Rb2, Rc, Rd, Re, and Rg1, and demonstrate different pharmacological activities [9,10]. In addition, the presence of rare ginsenosides, namely pentacyclic oleanane saponin Ro (3,28-O-bisdesmoside) [11] and ocotillol saponin F11 (24-R-pseudoginsenoside) [12], has been documented.

In the past few decades, studies have extensively explored the pharmacological potential of ginseng in drug development, demonstrating potent efficacies as an anticancer, antioxidative, antidiabetic, immune booster, and neuroprotective agent [9,13,14,15]. Among multiple pharmacological bioactivities, perhaps one of the most exciting biomedical applications of ginseng and ginsenosides focuses on cancer treatment [16]. Moreover, the phytoconstituents in ginseng, the triterpenoid ginsenosides, acidic polysaccharides, and phenols demonstrate good efficacies in countering different cancer types, displaying different molecular mechanisms [13,17] (Figure 1).

The development and regular intake of plant products prevent cancer occurrence and promote overall well-being in humans [18]. The increasing literature evidence has reported that ginsenosides Rc, Rd, Rb1, Rb2, Rh2, and others effectively counter multiple cancer cells by targeting different pathways and molecules; they also initiate autophagy, apoptosis via p53 pathway regulation, Reactive Oxygen Species (ROS) neutralization, normalization of the Nuclear factor kappa-light-chain-enhancer of activated B cells (NF-Kb) pathway, and regulation of *B-cell lymphoma 2* (Bcl-2) expression, and block inflammatory pathways, among other mechanisms [19]. Their rapid expansion in clinical trials is attributed to their therapeutic significance and the progress made in the development of ginseng-derived products and formulations [20,21]. However, less polarity, low permeability across membranes, and poor metabolism in the human body hamper the application of ginsenosides as drug candidates [22]. Despite the promising bioactivities of ginsenosides, a low absorption rate (<5%) in the gastrointestinal tract causes low bioavailability. One possible hypothesis suggests that ginsenoside interactions with cell membranes, ion channels, and receptors (intracellular) via transcriptional variations [23], cytotoxicity, and poor solubility hamper further clinical investigations [24,25], further complicated by efflux transporters that pump ginsenosides out of the cell [26,27]. To address these gaps, advanced biomolecular ginsenoside conjugation and drug delivery approaches aim to improve the bioavailability and membrane permeability of ginsenosides [28]. In drug development, drug delivery methods comprise an integral approach and are classified according to the following routes: intranasal, subcutaneous, pulmonary, intravenous, and transdermal. Each route has key advantages and limitations [29,30]. Multiple-drug delivery systems include solid–lipid particles, liposomes, niosomes and micelles, nanoparticle based-chitosan, inorganic nanoparticles (NPs), silica NPs, and metal-based NPs, amongst others [31]. The era of new medicine has witnessed considerable progress in drug delivery systems; however, feasible solutions are required to address bottlenecks in safety evaluation, adverse effects, fast elimination [32,33], regulatory guidelines, and manufacturing expense [34].

In the present era, green nanotechnology is gaining recognition to ensure sustainable development with emerging applications in bio-based therapeutics, bioremediation, and renewable energy. A significant rise is observed in the global green nanotechnology market, estimated to reach ~417.35 billion in 2030 [35]. One such area of profound development is herbal medicines, where nanotechnology has substantially contributed to improving the potential efficacies, stability, and bioavailability of drug-like molecules [36]. In addition, nano-herbal formulations improve controlled and targeted drug delivery to tissues; however, a safety assessment of formulations is required to prevent toxicity [37].

The emerging “nanotechnological era” has witnessed considerable progress in the phyto-mediated synthesis of nanoparticles with favorable implications, specifically in ginseng [38]. *P. ginseng*, followed by *P. notoginseng* and *P. quinquefolius,* is widely employed for different nanoparticle synthesis applications. While nanotechnological interventions are promising for enhancing ginsenoside properties, ongoing research suggests that nano-ginseng could address the limitations associated with ginsenoside bioavailability and cell membrane permeability and facilitate effective and targeted drug delivery [38]. The ongoing research on ginseng-based nanotherapeutics is promising for cancer treatment; however, further investigations are needed to validate these findings and address the associated bottlenecks. This article discusses the current advances in the development of ginseng-based nanotherapeutics, how the research gaps in ginseng pharmacokinetics can be addressed, translational achievements, and future perspectives/directions in the field.

## 2. Research Methodology: Bibliographic Source, Compilation, and Analysis

The literature review was conducted according to the Preferred Reporting Item for Systematic Reviews and Meta-Analysis (PRISMA) guidelines [39]. Scholarly databases, including Google Scholar [40], Pubmed [41], and Scopus [42], were searched for studies related to the *Panax* genus, phytoconstituents, pharmacological applications, herbal formulations, nanobiotechnological interventions in ginseng, and cancer treatment. Figure 2 provides a flowchart workflow for the studies screened, retrieved, and selected for the compilation of the literature review. The studies discussing the concepts and methodologies in this field were selected based on the following criteria: (a) research papers discussing pharmacological properties and applications of ginseng, (b) emerging concepts and investigations on ginseng metabolites for cancer treatment, and (c) nano-based ginseng formulations, their prospects, and limitations. Exclusion criteria comprised (a) no full-text found, (b) language other than English, and (c) other reasons to exclude them after abstract screening. Finally, 122 studies were referred for compilation and execution of the literature review.

## 3. Ginseng Formulations and Metabolites as Herbal Medicines: Authentication and Commercialization

The plants classified in the genus *Panax* have a traditional history of use in Chinese, Japanese, and Korean herbal medicine, as well as in Western countries. Research investigations have demonstrated the beneficial properties of ginseng and its bioactive constituents, ginsenosides, in treating multiple human disorders [17,43], often known as the ‘King of Herbs’. Studies have documented the use of ginseng since the 15th century, while Zhong Jing Zhang, an eminent scholar, first described its use in herbal preparations. Studies have documented the use of ginseng as a tonic for fatigue and sexual health, as well as for improved concentration [44]. Ginseng also has multi-faceted bioactivities. Worldwide, ginseng extracts and formulations are developed and marketed, suggesting its potential pharmacological functions in treating human ailments [20,45]. The commonly used species include *P. ginseng*, *P. quinquefolius*, *P. notoginseng*, *P. japonicus*, *P. pseudoginseng*, and *P. vietnamensis* [45]. Focusing on the commercial aspect, most ginseng products are derived from sustainable wild varieties, while the use of declining uncontrolled wild species is limited [46]. The quality and properties of the products are influenced by multiple factors, including geographical location, cultivar type, and growth conditions [47]. Wu et al. [48] showed that environmental factors (temperature in different seasons and soil phosphorous availability) considerably affect the accumulation of secondary metabolites and the quality of ginseng. In the study, plant and soil samples were collected from different geographical locations, and secondary metabolites were quantified by pathway analysis, the Mantel test, Pearson’s correlation, and random forest analysis. The results showed that the seasonal temperature changes and soil phosphorous availability were the stronger drivers of the secondary metabolites’ profile in ginseng, emphasizing the need for optimal ginseng cultivation for high-quality varieties.

Lee et al. [49] quantitatively analyzed dammarane-type saponins in different ginseng products (employing high-performance liquid chromatography with ultraviolet spectroscopy HPLC-UV) to optimize ginseng cultivation and increase the production of new ginsenoside analogs. Ginseng in the global market has a prominent place, with an economic worth of more than USD 2.1 billion [50]. The marketed ginseng products are comprised of different plant formulations, namely capsules, powders, gelcaps, teas, entire plant roots, etc. Some of the marketed ginseng extracts are North American Ginseng Extract (NAGE) from *P. quinquefolius* (total ginsenoside content adjusted to 10%) (Canadian Phytopharmaceuticals Corporation, Canada) and G115 from *P. ginseng* (total ginsenoside adjusted to 4%) (Pharmaton SA, Switzerland) [8], among others. The growing global market for ginseng-derived products is expanding substantially; however, substitution and adulteration are key bottlenecks, hampering the economic outcomes of different ginseng products in the market [51].

## 4. *Panax* Genus and Cancer Treatment: Understanding Recent Developments

Ginseng has been used as a herbal medication since time immemorial and is gaining recognition in the present scenario. The potent pharmacological efficacies of ginseng and ginsenosides in countering multiple human ailments are fascinating, defining interesting avenues in cancer therapeutics. Ginsenosides exhibit multiple anticancer mechanisms, including inducing apoptosis, hampering tumor cell proliferation, and restricting cell migration and invasion [52], verified in multiple research studies. In addition, studies have documented the inhibitory effect of ginsenosides on tumor cell proliferation. A case study showed that the invasion and proliferation of gastric cancer cells are hindered by ginsenoside Rh1 via targeting the Trans forming growth factor β/Suppressor of mother against decapentaplegic family proteins (TGF-β/Smad) pathway [53]. Moreover, the molecular mechanisms of ginsenoside Rh1 comprise greater production of ROS, caspase 3 cleavage, and Recombinant Activating Transcription Factor 4 (ATF4) increases [54]. Ginsenoside Rh2 exerts dual anticancer mechanisms via the upregulation of LncRNA STXBP5-AS1 expression and breast cancer cell inhibition [55], the restriction of non-small-cell lung cancer (NSCLC) progression, and the hampering of the Wingless/Integrated (Wnt) signaling pathway [56]. An inhibitory effect of ginsenoside Ro on carcinoma cells (hepatocellular) was observed [57]. Furthermore, multiple anticancer mechanisms of ginsenoside have been discussed: enhanced miR-150-3p expression, cell proliferation inhibition, and SRCIN1/Wnt pathway regulation in colorectal cancer by ginsenoside Rh2 [58]. Ginsenoside Rh3 has been documented to have potent anticancer effects on cancer cell lines; key effects include restricting the invasion and proliferation of colorectal cells, the downregulation of Deoxyribonucleic acid (DNA) replication proteins, and G1-phase cell cycle arrest by ginsenoside Rh3 [59]; the regulation of key proteins of the Hedgehog pathway and restricted proliferation and migration of epithelial–mesenchymal transformation (EMT) in lung cancer by 20(S)-ginsenoside Rg3 [60]; the anti-esophageal cancer cell activity of 20(S)-ginsenoside Rh2 [61]; and ginsenoside Rg3-mediated reduced circ_0003074 in miR-516b-5p/Karyopherin subunit alpha 4(KPNA4)-dependent expression and disease progression in osteosarcoma [62].

Worldwide, breast cancer is the leading cause of death in women [63], and ginsenoside Rh1 shows potent efficacy in cell cycle arrest (G1/S phase), inducing stress-mediated calcium accumulation, highlighting its positive therapeutic effects in triple-negative breast cancer [54]. In addition, ginsenoside Rh2 hampers the G1/S phase in the cell cycle and causes apoptosis [64]. In cancer cells, apoptosis leads to cell death; and multiple signal cascades are initiated in cell apoptosis by different ginsenosides: osteosarcoma cell apoptosis by ginsenoside Rg3 [65], CRC cell apoptosis by ginsenoside Rh3 [66], and cervical cell apoptosis [67], liver cancer apoptosis [68], and gastric cancer apoptosis [69] by ginsenoside Rg5. Literature studies have further discussed the favorable functions of ginsenoside analogs in addressing multiple cancer forms via different mechanisms: cell cycle arrest via suppression of cell cycle regulators, namely cyclin D_1_, cyclin-dependent kinase 4 (CDK_4_), and retinoblastoma protein (pRb), ROS-mediated cell cycle arrest [70], and the inhibition of cell invasion and migration by ginsenoside Rh4 [71], ginsenoside Rb2 [72], and ginsenoside Rg3 in colon cell migration and metastasis [73]. The study on tumor angiogenesis and anti-vascular therapy defines an exciting research field, and extensive research has delved into exploring the inhibition mechanisms of ginsenoside in angiogenesis. Interesting studies include the regulation of protein kinase B (Akt) and extracellular signal-regulated kinases (ERK) pathways and melanoma-induced angiogenesis by ginsenoside Rg3 [74], the hampering of tumor cell extravasation across the endothelium by restricting breast cancer cell invasion and the decrease in endothelial cell permeability by ginsenoside Rh1 [75], and the curbing of liver cancer via metastasis and restricted invasion by ginsenoside Rh2 and derivatives [76], among other studies. Progress in natural product-mediated cancer treatment approaches has provided significant insights and bridged the knowledge gap on ginsenosides and their molecular mechanisms in cancer [52,53,77,78].

## 5. Nanobiotechnological Advances in Ginseng: Addressing the Gaps in Therapeutics Development

The emerging field of nanobiotechnology in natural product-mediated drug discovery has witnessed remarkable breakthroughs. Nanobiotechnological advances in ginseng aim to address key limitations associated with bioavailability, half-life, and hydrophilicity. Studies in this direction have increasingly investigated nanobiotechnological applications in multiple species with promising outcomes. The bioactive ginsenosides are widely used and documented for their pharmacological properties, such as having anti-inflammatory, anticancer, antioxidant, neuroprotective, and cardioprotective effects, protecting renal functions, etc. [15,52,58,79]. The scientific and technological breakthroughs achieved in the isolation procedures and improved production techniques are instrumental in the clinical application of ginsenosides [80]. Key ginsenoside analogs, namely ginsenoside CK, ginsenoside Rg3, and ginsenoside Rd, have entered clinical trials and/or are commercially marketed for therapeutic purposes. However, major challenges in the development of ginsenoside as a drug molecule are its low permeability and bioavailability [81,82].

While plant-based drugs or herbal drugs are widely used and possess good efficacies, they suffer from challenges, including instability, low bioavailability, poor solubility, and lack of targeted delivery to tissues. Nano-herbal formulations integrate traditional herbal medicines with advanced technologies to address these limitations. They optimize solubility, expedite targeted drug delivery to tissues, and prevent degradation of the drug molecules [37], defining new avenues in drug development. In addition, nanoformulations possess better pharmacokinetics, solubility, and stability, causing increased absorption and longer circulation in the human body. Nanoencapsulation is a promising and effective method for improving antibacterial properties, bioavailability, and targeted drug administration [83]. Some significant studies on ginseng-based nanoencapsulation are as follows: Tang et al. [84] worked on improving the hydroxyl radical scavenging activity and bioavailability of 20(R)-ginsenoside Rg3 by loading it onto cellulose nanocrystals. Kim et al. [85] studied red ginseng-loaded chitosan NP fabrication to enhance antithrombotic function by employing fucoidan or polyglutamic acid as antithrombotic substances.

Bhavna et al. [86] developed an improved method of lignin nanoparticle (LNP) development for efficient delivery of ginseng. In their study, ginseng-loaded lignin nanoparticles were fabricated and characterized using scanning electron microscopy, X-ray diffraction, transmission electron microscopy, Fourier transform infrared spectroscopy (FT-IR), and other techniques. In addition, multiple parameters, including in vitro drug release, antioxidant activity, and drug loading, were evaluated, showing that ginseng-loaded NPs (GnLNPs) demonstrated 95% drug loading efficiency and good antimicrobial activity, providing a smart delivery platform for ginseng [86]. In other studies, 95% ginseng loading was selected based on the optimum size (200 µg/mL of ginseng yield) and polydispersity index (PDI). Furthermore, in vitro drug release from GnLNPs was evaluated: the cumulative drug release was 70% after 10 h (at pH 5.5) and gradually released slowly at pH 7.5 and pH 9.4, suggesting a key impact of the pH of the environment on drug release. Their study also determined the stability of GnLNPs and bare lignin NPs at regular intervals, stored at 4 °C (NPs surface potential, polydispersity, and size), and a slight difference in the mean particle diameter of LNPs (from ∼113 to 140 nm) and GnLNPs (from ∼137 to 169 nm) was observed, while the NPs’ PDI and zeta potential did not differ much after a month of storage. Under physiological conditions, the NPs did not precipitate or aggregate after a month, suggesting that the stability of GnLNPs was improved by hydrophobic interactions (hydrophobic parts of amphiphilic ginseng molecules and hydrophobic interactions between nonpolar lignin molecules) and that hydrophobic interactions of ginseng with water molecules improved NP stability [86].

Park et al. [87] investigated nanotechnological interventions in Korean black–red ginseng aiming to increase the solubility via reduced particle size. Nano-sized ginseng was developed via pulverization (particle size: 350 nm range), and increased concentrations of ginsenosides, lipids, minerals, carbohydrates, and proteins were detected. The results showed that nano-sized particles increased transport and absorption across the intestinal wall of mice, demonstrating promising results. To improve the therapeutic potential of ginseng in inflammatory bowel disease and mechanisms, Yang et al. [88] developed ginseng-based nanoparticles and studied their mechanisms in intercellular communication in inflammation. The research showed that ginseng-based nanoparticles act on Sequestosome 1 (p62)/Kelch Like ECH Associated Protein 1 (Keap1)/Nuclear factor erythroid2-related factor 2 (Nrf2) (p62/Keap1/Nrf2) and Toll-like receptor 4 (TLR4)/mitogen-activated protein kinase (MAPK) (TLR4/MAPK) pathways, scavenge ROS from intestinal epithelial cells and immune cells, hinder pro-inflammatory factors’ expression, and promote tissue repair, highlighting their therapeutic mechanisms in inflammatory bowel disease. Furthermore, Kim et al. [89] researched the anti-glioma function of ginseng-derived NPs to address challenges with limited ginseng permeability across the blood–brain barrier (BBB) and develop glioma treatment. In their study, the researchers developed ginseng-derived exosome-like NPs and studied their antitumor response and restriction of tumor progression. Both in vivo and in vitro studies showed the positive function of ginseng-derived NPs as glioma therapeutics for further development. Studies have also discussed the prominent activity of *P. notoginseng*-based NPs for the treatment of ischemic stroke [90]. A study investigated the development of *P. notoginseng*-based exosome-like NPs and reported that these NPs can penetrate the brain (without any changes) and ameliorate cerebral infarct volume. In addition, it maintains BBB integrity and attenuates Cerebral ischemia/reperfusion (CI/R) injury, displaying positive implications [90]. Although ginseng-based NPs are widely investigated for their therapeutic functions, further research is essential to understand the molecular mechanisms, address knowledge gaps, and facilitate research progress.

## 6. State-of-the-Art Progress in the Development of Ginseng-Based Cancer Nanotherapeutics

Worldwide, a notable increase has been witnessed in the prevalence of multiple cancer types, highlighting an urgent requirement for improved treatment procedures. Current chemotherapies are hindered by drug resistance, adverse effects, and damage to tissues [91], although they are advantageous in selectively targeting proliferating cancer cells. Extensive research investigations aim to develop new methods with minimum adverse effects and enhanced chemotherapeutic efficacy [92]. Nanotechnology-based drug delivery systems represent an innovative approach and utilize nanoscale drug delivery platforms that co-integrate phytochemicals with chemotherapeutic drugs to restrict the growth of cancer cells in preclinical studies and enhance therapeutic efficiency. Phytochemicals are plant constituents that are well documented for their anticancer activities and modulation of metabolic pathways associated with cancer progression [93]. These natural compounds exhibit diverse anticancer mechanisms, namely restricted cell proliferation, P-glycoprotein inhibition, apoptosis, and cancer cell migration, among others. Phytochemical-mediated modulation of multiple pathways and fewer side effects are key to developing co-combinational cancer therapies [92,93].

The therapeutic effect of ginseng is well known in multiple cancer types, including liver, colon, breast, skin, and lung cancer. In chemotherapies, ginseng improves the efficiency of chemotherapeutic drugs, decreases fatigue, and improves hair growth, acting as a chemosensitizer drug [94,95,96]. In another clinical study, the antitumor activity of paclitaxel and epirubicin drugs was enhanced by ginseng supplementation as a result of increased accumulation of *Bak* and *Bax* in mitochondria and cell death. Korean red ginseng improves hair growth in patients undergoing cancer chemotherapy [97]. Several other functional mechanisms of ginsenosides in cancer prevention and treatment have been discussed [98,99,100,101,102,103,104].

In the present decade, the potential prospects of ginseng-based nanotherapeutics are immense, emerging as therapeutic innovations on the global platform (Figure 3). The present therapeutic interventions aim to address the existing challenges associated with ginsenosides, poor oral bioavailability, cytotoxicity, and low solubility [105,106] via a conjugated ginsenoside-based drug delivery strategy [107]. Advances in nanodrug delivery carriers are projected to promote drug-like properties via better membrane penetration, solubility increases, enhanced anticancer activity, etc. [16]. Mathiyalagan et al. worked on ginsenoside CK and PPD (poor solubility) to improve delivery by using hydrophilic ginsenoside conjugates [106]. The conjugate developed comprised ginsenoside C-3 position of the OH group and was covalently linked to the amine group of hydrophilic glycol chitosan via linkage stemming from carbodiimide coupling reactions. The study showed a more rapid release of ginsenoside in acidic conditions (pH 5.0), like in the tumor environment, compared to pH 7.4 in in vitro studies. In HT29 cancer cells, polymer ginsenoside conjugates demonstrated increased cytotoxicity and promoted cell viability better than ginsenoside CK alone and hindered lipopolysaccharide (LPS)-induced production of nitric oxide [108]. Li et al. (2010) showed the potent cytotoxic and apoptotic activity of polyethylene glycol–poly lactic acid-co-glycolic acid-ginsenoside Rg3 (PEG–PLGA–Rg3) nanoparticles, while hybrid nano-complex NPs with better uptake and serum stability led to the longer circulation of (S)-Rg3 in in vivo studies [109]. Yang et al. (2014) developed an albumin-based nanocapsule loaded with ginsenoside Rg3, demonstrating higher efficiency [110]. The combined effects of Rg3 chemotherapy and hyperthermia (nanosphere-induced magnetic fluid) were highly effective in the apoptosis of HeLa cervical cancer cells in vitro [110].

The high-value medicinal plants from the *Panax* genus and their role in NP biosynthesis have gained key scientific recognition [25,38,111], while *P. ginseng*, *P. quinquifolius*, and *P. notoginseng* are employed as the representative species. Phytochemical-mediated NP synthesis (biological synthesis) influences the therapeutic efficiency of the NPs. In ginseng, phytochemicals act as reducing agents to form nano-sized particles and stabilize the coating on metal NPs [112], demonstrating distinct advantages. In the thematic literature, Wang et al. (2021) highlighted the emerging significance of ginsenosides as nanocarriers and bifunctional drugs for improved antitumor functions [17]. For precise cancer treatment, ginsenoside-based nano-delivery systems and technologies are documented: micelles, microemulsions, polymer nanoparticles, metals, and inorganic NPs [17]. For precise delivery and maximum antitumor activity, multiple targeted delivery systems are designed, exhibiting improved functions (barrier permeability and immune regulation) and defining a path forward from laboratory to clinical applications. Dai et al. [111] provided comprehensive insights to develop ginsenoside NPs as drug delivery systems. The study showed that ginsenoside Rb1 conjugated with anticancer drugs (dihydroartemisinin, betulinic acid, and hydroxycamptothecine) demonstrates better anticancer efficacies, improved pharmacokinetics, a remarkable prolonged clearance, and free drug levels of ~15% (RB1 nanoparticles) in mice (compared to only ~5% for ginsenosides), as well as tumor selectivity in mice models, defining novel progress in ginseng biotechnology [111].

Singh et al. emphasized the pharmacological value of ginseng-mediated synthesis of silver and gold NPs (fresh ginseng leaves), namely anticancer activity in A549 cancer cells, non-cytotoxicity in human keratinocytes, and anti-inflammatory activity in LPS-induced RAW264.7 macrophages [113]. The ginseng-derived metal NPs inhibited cell viability and oxidative stress in cancer cells without causing damage to normal cells, showing potent activity in human hepatoblastoma cell line (HepG2), Michigan Cancer Foundation-7 (MCF7), and adenocarcinomic human alveolar basal epithelial cells (A549) cancer cell lines. Castro-Aceituno et al. [103] synthesized a novel silver NP from *P. ginseng,* evaluated its oxidative and cytotoxic potential against A549, MCF7, and HepG2 cancer cell lines, and further studied the effect of silver NPs on apoptosis and cell migration and their anticancer mechanism. The results showed that silver NPs demonstrated anticancer activity against the A549 cell line via regulation of MAPK/p53 and epidermal growth factor receptor (EGFR)/p38 pathways [103]. Alinaghi et al. [114] reported the biosynthesis of platinum and palladium NPs using ginseng extracts and characterized NPs through Scanning Electron Microscopy (SEM), Transmission Electron Microscopy (TEM), Dynamic Light Scattering (DLS), and FT-IR spectroscopy techniques. The results showed the potent anticancer activity of the metal NPs against colon cancer cell lines via an MTT assay. Matsumura and Maeda [115] developed a nano-ginseng delivery system: Rb1/PPD NPs were fabricated with PPD and Rb1, ginseng (20S)-protopanaxadiol-type compounds. Studies conducted in vivo showed that the nano-delivery system (with particle size 120 nm) showed better efficiency due to enhanced permeability. The study showed that a novel nano-formulation of ginseng controls particle size and distribution, good biocompatibility, and anticancer activity, defining promising outcomes. Wang et al. [116] highlighted the development of ginseng-based carbon dots (GCDs) for the inhibition of the invasion and development of malignant tumors and the ability to kill cancer cells. In the study, GCDs were synthesized, purified, and tested against squamous cancer cell lines. The growth of Squamous cell carcinoma (SCC-25), adenosquamous carcinoma cell line (Cal-27), and Squamous cell carcinoma (SCC-7) cancer cells was inhibited by GCDs at > 250–300 μg/mL concentrations. Moreover, immunofluorescence revealed the improved permeability of GCDs across cells, while transcriptome studies showed gene expression profiles and mRNA expression in ferroptosis pathways. In vivo studies showed that GCDs improved CD4^+^ T-cell infiltration in the tumor and decreased cell invasion and migration. The study provided key insights into how GCDs could address key therapeutic challenges in ginseng drug development. Cao et al. [117] reported the isolation and characterization of ginseng-derived NPs from *P. ginseng*. Further, the study discussed the use of ginseng-derived NPs as an immunopotentiator for M2-polarized macrophage modulation that led to an antitumor response via MyD88 and TLR4 signaling. In addition, ginseng-derived NPs act as an immunomodulator in mammalian immune response, establishing the emerging concept of nano-drugs in cancer chemotherapies [117]. Furthermore, a research investigation into how ginseng-derived NPs reprogram macrophages to regulate the release of arginase-1 and enhance T-cell response was presented by Lv et al. [118]. The study established the molecular mechanism of ginseng-derived NPs and their promising implications in the tumor microenvironment.

## 7. Future Directions

The global rise in disease burden is a major crisis for economies, particularly the rising frequency of multiple cancer types, a leading cause of mortality. While the field of cancer disease management has witnessed considerable progress, current chemotherapies are hindered by challenges in tissue damage, drug resistance, and side effects of chemotherapeutic drugs. The nanotherapeutic intervention has opened new avenues in cancer biology, with the phytochemical-mediated development of nanoconjugates achieving key translational success. The plants of the *Panax* genus are globally known and documented for their pharmacological benefits, demonstrating potent efficacies in animal models; however, bottlenecks in ginseng cytotoxicity, permeability, stability, and bioavailability hamper clinical research. Advanced nanobiotechnological techniques, e.g., the development of ginsenoside nanoconjugates and nanocarriers as a delivery system (proteins, micelles, liposomes, and polymers), improve water solubility, minimize cytotoxicity, and show good anticancer potential in clinical studies. The scope of ginseng-based nanotherapeutics is promising, with the ability of ginseng-derived NPs to penetrate the blood–brain barrier and modulate the tumor microenvironment, demonstrating the potential to regulate tumor-associated macrophages (TAMs) and restrict glioma progression. Another key achievement in ginseng nanobiotechnology focuses on the development of pH-responsive nanocarriers for triple-negative breast cancer (TNBC). Zuo et al. [119] designed a ginsenoside-based nanodrug (composed of ginsenosides Rb1 and Rg3) using the nano-reprecipitation method. The Rg3-Rb1 NPs demonstrated strong anti-invasive and antitumor effects on TNBC in vitro, opening exciting possibilities of engineered ginsenoside nanodrugs in TNBC therapy. However, some translational hurdles hamper the progress of nanotherapeutics in clinical applications in terms of regulatory guidelines. For the clinical translation of nanotherapeutics, some criteria must be considered in terms of early-stage product development, including the considered drug’s quality features, safety, bioactivity, the analytical methods used for product characterization, the parameters of the process, and sufficient pre-clinical models for validation. Some other concerns, namely batch-to-batch consistency and scaling up the process, need to be monitored to facilitate a reproducible manufacturing process and compliance with the regulatory guidelines. The progress and translational success achieved in ginseng-based nanotherapeutics is commendable and defines promising outcomes in the development of advanced cancer therapies.

## Figures and Tables

**Figure 1 cimb-47-00250-f001:**
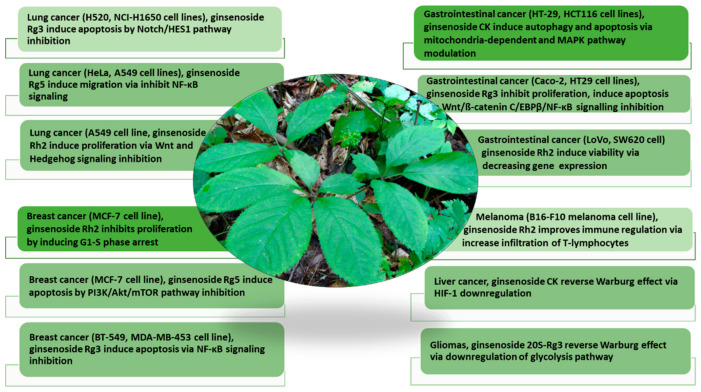
The pharmacological activities of ginsenosides in countering multiple cancers (lung cancer, breast cancer, gastrointestinal cancer, melanoma, liver cancer, and gliomas) and their molecular mechanisms (modified from Wang et al. (2021) [17]). The anticancer mechanisms of ginsenoside analogs on different cancer cell types via targeting different metabolic pathways are discussed.

**Figure 2 cimb-47-00250-f002:**
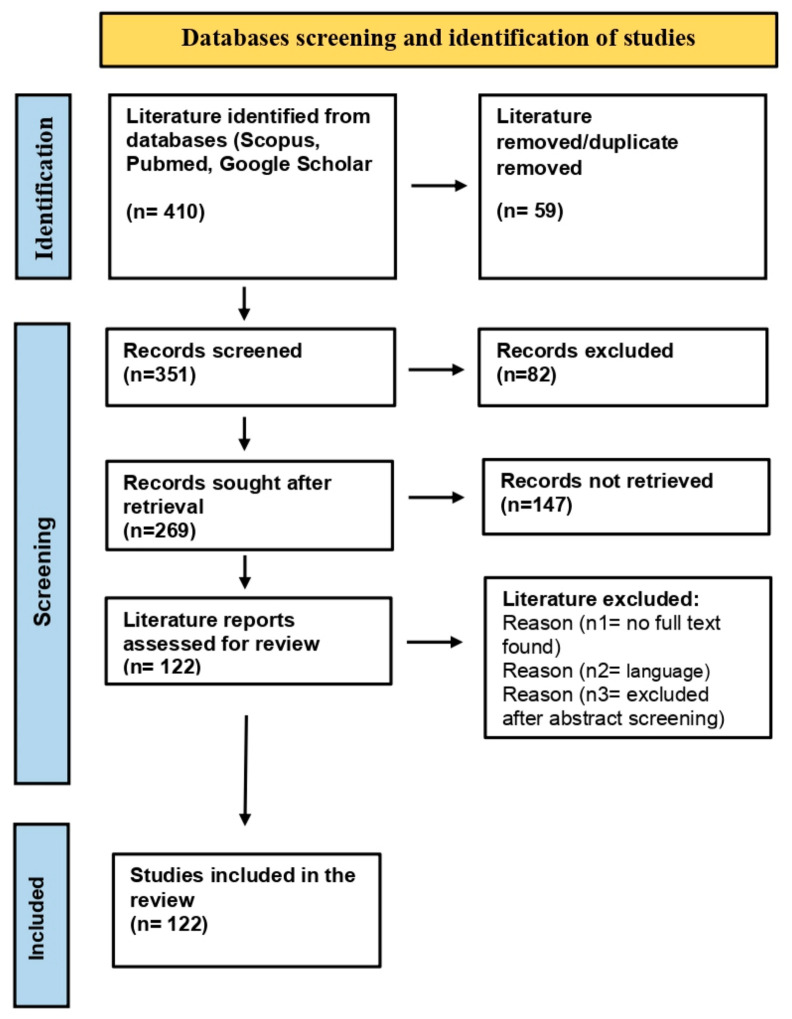
Flowchart workflow for studies screened, retrieved, and included in literature review.

**Figure 3 cimb-47-00250-f003:**
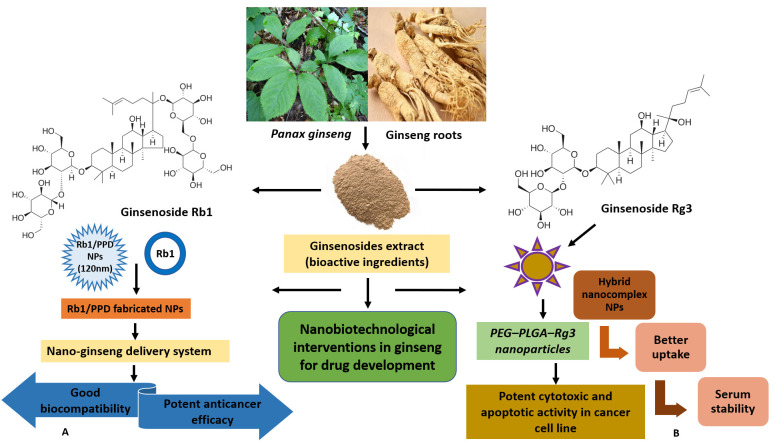
Advanced nanobiotechnological approaches in the development of ginseng-based therapeutics. *Panax ginseng* and its bioactive constituents (ginsenosides) are the key components for the development of nanoconjugates. (**A**) In the study, ginsenoside Rb1/PPD-fabricated NPs were synthesized as the nano-ginseng delivery system, and they showed good biocompatibility and potent anticancer efficacy. (**B**) In the study, PEG–PLGA–Rg3 nanoparticles (polymeric NPs) exhibited significant cytotoxicity and apoptosis in C6 glioma cells, stability, and better uptake in pharmacokinetic analysis.

## Data Availability

Not applicable.

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
