# Peer review of "Ginseng-Based Nanotherapeutics in Cancer Treatment: State-of-the-Art Progress, Tackling Gaps, and Translational Achievements"

_cimb, 2025, doi:10.3390/cimb47040250_

Round 1

Reviewer 1 Report

Comments and Suggestions for Authors

Overall evaluation

This review comprehensively addresses the potential of ginseng-based nanotherapeutics in cancer treatment, covering pharmacological mechanisms, nanotechnological advancements, and translational challenges. The scope is ambitious, and the integration of green nanotechnology with traditional herbal medicine aligns well with current research trends. However, the manuscript suffers from insufficient methodological rigor, inconsistent data presentation, and structural ambiguities. While the topic is timely, significant revisions are required to enhance clarity, support claims with robust evidence, and align with academic standards.

Specific problems and suggestions for improvement

  1. Abstract (Page 1, Lines 10-15)

The abstract lacks quantifiable outcomes (e.g., bioavailability improvement percentages) and fails to distinguish nano-delivery systems from traditional methods.

  1. Introduction (Page 1, Lines 25-27)

The link between Sustainable Development Goals (SDGs) and ginseng research is vague.

  1. Research Methodology (Page 3, Section 2)

Add a flowchart or table summarizing the PRISMA-style selection process, including the number of papers screened, excluded, and analyzed.

  1. Commercialization Section (Page 4, Lines 8-10)

Claims about geographic factors affecting ginseng quality lack empirical support.

  1. Nanobiotechnological Advances (Page 5, Lines 20-22)

The statement "95% drug loading efficiency" lacks context (e.g., release kinetics, stability under physiological conditions).

  1. Figure 1 and 2 (Pages 2 and 7)

Figures are referenced but not included in the submitted manuscript.

Ensure figures are uploaded and described in detail (e.g., label key pathways in Figure 1, annotate nanoparticle structures in Figure 2).

  1. References

Inconsistent formatting (e.g., journal names italicized inconsistently, missing DOIs).

  1. Language and Style (Page 6, Lines 15-18)

Awkward phrasing (e.g., "pretty much addressed by a conjugated strategy") and grammatical errors (e.g., "aim for effective and targeted drug delivery").

  1. 9. Future Directions (Page 8, Section 7)

Prioritize actionable goals (e.g., "pH-responsive nanocarriers for triple-negative breast cancer" or "clinical trials evaluating GCDs in glioblastoma").

Comments on the Quality of English Language

The English could be improved to more clearly express the research.

Author Response

This review comprehensively addresses the potential of ginseng-based nanotherapeutics in cancer treatment, covering pharmacological mechanisms, nanotechnological advancements, and translational challenges. The scope is ambitious, and the integration of green nanotechnology with traditional herbal medicine aligns well with current research trends. However, the manuscript suffers from insufficient methodological rigor, inconsistent data presentation, and structural ambiguities. While the topic is timely, significant revisions are required to enhance clarity, support claims with robust evidence, and align with academic standards.

Specific problems and suggestions for improvement

  1. Abstract (Page 1, Lines 10-15)

The abstract lacks quantifiable outcomes (e.g., bioavailability improvement percentages) and fails to distinguish nano-delivery systems from traditional methods.

Reply: The authors thank the esteemed reviewer. The abstract was revised to include quantifiable results, the suggestions were included in the revised manuscript,

  1. Introduction (Page 1, Lines 25-27)

The link between Sustainable Development Goals (SDGs) and ginseng research is vague.

Reply: Thank you for the suggestion. In the present era, herbal medicines are gaining recognition as potent and safe alternatives for the treatment of human diseases. However, there are key factors hampering the progress in herbal medicine research, and these comprise: regulation, industrialization, biopiracy, inadequate infrastructure, overuse and irrational use of medicinal herbs, and loss of biodiversity. With key phytomolecules documented for their therapeutic potential, herbal medicines have a promising potential to support the SDGs of the United Nations. While ginseng is not directly researched for its role in promoting sustainable development, further research is necessary to investigate bio-based resources for healthcare, environment, and ecological subsistence.

  1. Research Methodology (Page 3, Section 2)

Add a flowchart or table summarizing the PRISMA-style selection process, including the number of papers screened, excluded, and analyzed.

Reply: A flowchart workflow based on a PRISMA-style selection process is included in the revised manuscript. In addition, the research methodology was revised as per suggestions.

  1. Commercialization Section (Page 4, Lines 8-10)

Claims about geographic factors affecting ginseng quality lack empirical support.

Reply: Thank you for your queries. A short section was added discussing the experimental findings on how the seasonal temperature changes and available phosphorous significantly affect the secondary metabolite profile and quality of ginseng (Wu and coworkers 2024). Please refer to the revised manuscript.

  1. Nanobiotechnological Advances (Page 5, Lines 20-22)

The statement "95% drug loading efficiency" lacks context (e.g., release kinetics, stability under physiological conditions).

Reply: We agree with the insightful comment. The literature study on "95% drug loading efficiency" was rewritten and revised to include the multiple contexts of physiological stability and release kinetics parameters of ginsenoside-loaded NPs (GnLNPs), discussed under the section “Nanobiotechnological advances in ginseng”; please see the highlighted changes.

  1. Figure 1 and 2 (Pages 2 and 7)

Figures are referenced but not included in the submitted manuscript.

Ensure figures are uploaded and described in detail (e.g., label key pathways in Figure 1, annotate nanoparticle structures in Figure 2).

Reply: The figures (figure 1 and figure 2) were included in the manuscript. It was further discussed in detail in the revised manuscript.

  1. References

Inconsistent formatting (e.g., journal names italicized inconsistently, missing DOIs).

Reply: The formatting was revised and corrected. Missing DOIs were included.

  1. Language and Style (Page 6, Lines 15-18)

Awkward phrasing (e.g., "pretty much addressed by a conjugated strategy") and grammatical errors (e.g., "aim for effective and targeted drug delivery").

Reply: Thank you for the suggestions. The manuscript was extensively revised to remove grammatical errors and inappropriate phrasing. Please refer to the revised manuscript for the changes.

  1. 9. Future Directions (Page 8, Section 7)

Prioritize actionable goals (e.g., "pH-responsive nanocarriers for triple-negative breast cancer" or "clinical trials evaluating GCDs in glioblastoma").

Reply: The authors agree with the key suggestion. The section “future directions” has been revised and rewritten to prioritize action goals and prospects of ginseng-based nanotherapeutics. The changes are highlighted in the revised manuscript.

The manuscript was extensively revised to improve the quality of the English Language.

Reviewer 2 Report

Comments and Suggestions for Authors

Minor Comments:

Lines 46–50: What are the key structural characteristics that differentiate protopanaxadiol (PPD) and protopanaxatriol (PPT) classes of ginsenosides, and how do these structural differences influence their pharmacological activity?

Lines 93–102: Through which molecular pathways do ginsenosides mediate apoptosis in cancer cells, and what are the key signaling cascades involved?

Lines 105–108: How does the limited gastrointestinal absorption (<5%) of ginsenosides affect their pharmacokinetic profile and subsequent clinical efficacy in cancer treatment?

Lines 110–112: What role do ATP-binding cassette (ABC) efflux transporters play in reducing intracellular accumulation of ginsenosides, and how does this impact their therapeutic potential?

Lines 115–120: In what ways can green nanotechnology-based drug delivery systems overcome the pharmacokinetic limitations associated with ginsenosides?

Lines 120–125: What are the comparative advantages of nanoformulated herbal drugs over conventional plant extracts in terms of targeted delivery, stability, and bioavailability?

Lines 180–183: How does ginsenoside Rh1 modulate the TGF-β/Smad signaling axis to inhibit gastric cancer cell proliferation and promote apoptotic signaling?

Lines 185–187: What is the mechanistic basis for the anticancer effect of ginsenoside Rh2 through the regulation of long non-coding RNA STXBP5-AS1 in breast cancer cells?

Lines 316–323: What preclinical evidence supports the use of ginseng-derived nanoparticles for effective blood-brain barrier penetration and glioma therapy?

Lines 359–364: How do ginseng-based carbon dots (GCDs) exert anti-tumor effects at the cellular and molecular levels in squamous cell carcinoma models?

Lines 367–373: How do ginseng-derived nanoparticles act as immunomodulators by altering macrophage polarization and activating TLR4 and MyD88 signaling pathways?

Lines 260–265: What is the role of nanoencapsulation in enhancing the pharmacological efficacy, stability, and controlled release of ginsenosides such as Rg3?

Lines 248–252: What are the primary challenges associated with the cytotoxicity, solubility, and stability of ginsenoside formulations, and how can nanoconjugation strategies mitigate these issues?

Lines 380–385: What translational hurdles currently limit the clinical application of ginsenoside-based nanotherapeutics, and what future research directions could address these gaps?

Author Response

Lines 46–50: What are the key structural characteristics that differentiate protopanaxadiol (PPD) and protopanaxatriol (PPT) classes of ginsenosides, and how do these structural differences influence their pharmacological activity?

Reply: According to the structural characteristics of steroidal saponins, ginsenosides are mainly classified as a) protopanaxadiol-type saponins (PDS, Rb1, Rb2, Rd, Rc, Rh2, CK), and PPD) and protopanaxatriol-type saponins (PTS, mainly including Re, R1, Rg1, Rh1, Rf, and PPT).

The key structural characteristics between PPD and PPT-type saponins:

PPD-type ginsenosides contain sugar moieties attached to the β-OH at the C-3 and/or C-20 position, whereas PPT-type ginsenosides include sugar moieties attached to the α-OH at C-6 and/or the β-OH at C-20.

The differences between PDS and PTS structure result in the differences in pharmacological activities. Moreover, ginsenosides have different properties depending on the attached sugar moieties.

The antioxidant activity of ginsenoside is documented in both in vivo and in vitro studies. For example, a study showed that the antioxidative effect of PDS was stronger than that of PTS, while the antioxidation of the Z-configuration was better than the E-configuration of pseudo-Rg2, pseudo-Rh2, pseudo-Rg3, and pseudo-Rh1 epimers.

In addition, the antioxidative activity of ginsenosides with low-sugar chains was higher than ginsenosides with high-sugar chains in PDS.

Lines 93–102: Through which molecular pathways do ginsenosides mediate apoptosis in cancer cells, and what are the key signaling cascades involved?

Reply: Ginsenoside mediates apoptosis in different cancer cells via multiple pathways.

Multiple studies have documented that ginsenosides like Rb1, Rb2, Rb3, Rc, Rd, Rg3, Rh2 etc. can prevent and treat cancer by targeting different pathways and molecules by induction of autophagy, neutralizing ROS, induction of cancerous cell death by controlling the p53 pathway, modulation of miRNAs by decreasing Smad2 expression, regulating Bcl-2 expression by normalizing the NF-Kb pathway, inhibition of inflammatory pathways by decreasing the production of cytokines like IL-8, causing cell cycle arrest by restricting cyclin E1 and CDC2, and induction of apoptosis during malignancy by decreasing β-catenin levels etc.

In breast cancer, ginsenoside Rg3 induces apoptosis via NF-κB signaling inhibition. In gastrointestinal cancer, ginsenoside Rg3 inhibits proliferation, induces apoptosis via Wnt/ß-catenin C/EBPβ/NF-κB signalling pathway inhibition, and in lung cancer, ginsenoside Rg3 induces apoptosis by inhibition of Notch/HES1 pathway. Moreover, inhibition of NF-κB

signaling is another mechanism by which ginsenoside Rg5 induces cell migration.

Furthermore, Ginsenoside Rg3 (G-Rg3) effectively suppressed human colorectal cancer (CRC) cell proliferation by inhibiting the transactivation of CCAAT/enhancer binding protein (C/EBP) and nuclear factor kappa-B (NF-κB), and the interaction of C/EBPβ with p65.

While, Ginsenoside Rh1 (G-Rh1) executed anti-metastatic activity through the inhibition of matrix metalloproteinase-1 (MMP-1) transcriptional activity and suppressing the expression and stability of the activator protein-1 (AP-1) dimer, c-fos proto-oncogene protein (c-Fos) and c-Jun by abrogating c-Jun N-terminal kinases (JNK) and extracellular signal-regulated kinase ½ (ERK1/2) activation in HepG2 cells

Lines 105–108: How does the limited gastrointestinal absorption (<5%) of ginsenosides affect their pharmacokinetic profile and subsequent clinical efficacy in cancer treatment?

Reply: Ginsenosides are high molecular weight secondary metabolites (usually more than 500 daltons) and have low bioavailability due to both metabolism and poor intestinal absorption. The factors limit their further applications.

The pharmacokinetics of a drug molecule (drug absorption, distribution, metabolism, excretion, and how the body affects the drug) are key factors in drug development. While drug absorption is very important in the human body, most drugs are administered orally, therefore, gastrointestinal (GI) drug absorption is considered. Ginsenosides show poor gastrointestinal absorption (<5%), and these factors lead to poor bioavailability and a poor pharmacokinetic profile, hampering its further development for cancer treatment.

Lines 110–112: What role do ATP-binding cassette (ABC) efflux transporters play in reducing intracellular accumulation of ginsenosides, and how does this impact their therapeutic potential?

Reply: The study investigated the mechanism of why ginsenoside showed poor absorption in the intestine.  The major absorption mechanism comprised carrier-mediated transport, Rh2 was a substrate for ABC transporters. The poor permeability of ginsenoside Rh2 and ABC-transporter-mediated efflux resulted in Rh2 poor absorption. It results in low bioavailability and a poor pharmacokinetics profile, thereby limiting the chances of further drug development, although it may show good bioactivities.

Lines 115–120: In what ways can green nanotechnology-based drug delivery systems overcome the pharmacokinetic limitations associated with ginsenosides?

Reply: Green nanotechnology-based development of drug-delivery systems has the potential to address the limitations associated with ginsenosides pharmacokinetics, as described in multiple research studies.

Multiple types of drug-delivery systems include: solid–lipid particles, liposomes, niosomes and micelles, nanoparticle-based-chitosan, inorganic nanoparticles (NPs), silica NPs, and metal-based NPs. The advanced biomolecular ginsenoside conjugation and drug delivery approaches aim to improve the bioavailability and membrane permeability of ginsenosides. For example: Bhavana et al. developed an improved method of lignin nanoparticle (NP) development for efficient delivery of ginseng. In the study, ginseng-loaded lignin nanoparticles were fabricated and characterized using scanning electron microscopy, X-ray diffraction, transmission electron microscopy, Fourier transform IR spectroscopy, and other techniques. In addition, multiple parameters, including in vitro drug release, antioxidant activity, and drug loading, were evaluated, showing that ginseng-loaded NPs demonstrated 95% drug loading efficiency and good antimicrobial activity, providing a smart delivery platform for ginseng.

Dai et al. [113] provided comprehensive insights and progress to develop ginsenosides NPs as drug delivery systems. The study showed that ginsenoside Rb1 conjugated with anticancer drugs (dihydroartemisinin, betulinic acid, and hydroxycamptothecine) demonstrate better anticancer efficacies, improved pharmacokinetics and remarkable prolonged clearance and free drug levels of ~15% (RB1 nanoparticles) in mice (compared to only ~5% for ginsenosides) and tumor selectivity in mice models, defining novel progress in ginseng biotechnology.

Lines 120–125: What are the comparative advantages of nanoformulated herbal drugs over conventional plant extracts in terms of targeted delivery, stability, and bioavailability?

Reply:

 Medicinal plants and their extract/phytomolecules are extensively recognized as potential alternatives to treat human ailments. While plant-based drugs or herbal drugs are widely used and possess good efficacies, these suffer from challenges including stability, bioavailability, poor solubility, and targeted delivery to the tissues. Nano-herbal formulations are advantageous and enhance the efficacy and delivery of therapeutic agents through nanotechnology, facilitating the integration of traditional herbal medicine and technological advances. The nanoformulations address the limitations of conventional herbal medicines by optimizing their solubility, protecting active compounds from speedy degradation, and targeting this drug to the part of the human body that requires it and nano-herbal formulations are gaining significant attention in drug development.

In addition, Nanoherbal formulations have improved pharmacokinetics, which are the main characteristics of these formulations, including better solubility and stability; therefore, the drug is better absorbed and stays longer in the body. These formulations are easier to deliver to specific targets; thus, the drug is released only in the affected areas. 

Lines 180–183: How does ginsenoside Rh1 modulate the TGF-β/Smad signaling axis to inhibit gastric cancer cell proliferation and promote apoptotic signaling?

Reply: Yang et al. (2022) studied the anticancer mechanisms of ginsenoside Rh1 in gastric cancer. The human gastric cancer cells were treated with different concentrations of ginsenoside Rh1 for 48 hrs, and multiple processes (proliferation, migration, invasion, and apoptosis) were assessed by MTT, scratch test, Transwell, and TUNEL tests. The results showed that ginsenoside Rh1 inhibited GC proliferation and induced tumour cell apoptosis. Mechanistically, Ginsenoside Rh1 reduced TGF-β1 and TGF-β2 levels and Smad2 and Smad3 phosphorylation levels. The findings highlighted that ginsenoside Rh1 inhibited GC cell growth and tumor growth in xenograft tumor models via inhibition of the TGF-β/Smad pathway.

Lines 185–187: What is the mechanistic basis for the anticancer effect of ginsenoside Rh2 through the regulation of long non-coding RNA STXBP5-AS1 in breast cancer cells?

Reply: Ginsenoside Rh2 was studied for its role in the suppression of breast cancer proliferation by Park and coworkers (2021). The long noncoding RNAs and microRNAs, regulated by ginsenoside Rh2 in cancer cells, were investigated.  The long noncoding RNAs (that promote methylation levels) were altered by ginsenoside Rh2. Apoptosis was decreased due to STXBP5-AS1 inhibition, but the growth of MCF-7 cells was stimulated showing tumor-suppression by the lncRNA. MiR-4425 was downregulated by STXBP5-AS1 as well as by Rh2. In contrast to STXBP5-AS1, miR-4425 showed pro-proliferation activity by inducing a decrease in apoptosis but increased growth of the MCF-7 cells. The results concluded that ginsenoside Rh2 controls the STXBP5-AS1/miR-4425/RNF217 axis to suppress breast cancer cell growth.

Lines 316–323: What preclinical evidence supports the use of ginseng-derived nanoparticles for effective blood-brain barrier penetration and glioma therapy?

Reply: Kim et al. [90] researched the anti-glioma function of ginseng-derived NPs to address challenges with limited ginseng permeability across the blood-brain barrier (BBB) and develop glioma treatment. In the study, the researchers developed ginseng-derived exosome-like NPs and studied the anti-tumor response and restricted tumor progression. Both the in vivo and in vitro studies showed the positive function of ginseng-derived NPs as glioma therapeutics for further development.

Lines 359–364: How do ginseng-based carbon dots (GCDs) exert anti-tumor effects at the cellular and molecular levels in squamous cell carcinoma models?

Reply: Ginseng carbon dots (GCDs) were developed for targeting squamous cell carcinoma and inhibiting the invasion and development of cancer cells. GCDs showed improved permeability to cross membranes and exert antitumor effects by improved CD4+ T cell infiltration in the tumor and decreased cell invasion and migration. Moreover, GCDs induced the ferroptosis of cancer cells, as shown by decreased GPX-4 and increased COX-2 expression. GCDs also decreased cell invasion and migration. In vivo, GCDs decreased tumor growth without apparent organ toxicity and promoted CD4+ T cell infiltration in the tumor.

Lines 367–373: How do ginseng-derived nanoparticles act as immunomodulators by altering macrophage polarization and activating TLR4 and MyD88 signaling pathways?

Reply: In order to understand how plant-derived extracellular vesicles (EVs) mediate interspecies communication, Cao and coworkers (2019) studied whether ginseng-based EVs modulate M2-like polarization (Tumor-associated macrophages (TAMs) have different polarization states between tumoricidal M1 phenotype and tumor-supportive M2 phenotypes, with a lower M1/M2 ratio correlating with tumor growth, angiogenesis, and invasion) and its role to promote cancer immunotherapy.

In the study, ginseng-derived nanoparticles were isolated and characterized from ginseng. Ginseng-derived nanoparticles were used as immunomodulators for the alteration of macrophage polarization, and it caused polarization of M2 to M1 via TLR4 and MyD88 mediating signaling, ROS production and caused apoptosis of melanoma cells. Moreover, the presence of proteins and ceramide lipids assisted in macrophage polarization via TLR4 activation. An increase in M1 macrophages in tumor cells was detected, and ginseng-derived nanoparticles suppressed melanoma growth in tumor cells in mice.

Lines 260–265: What is the role of nanoencapsulation in enhancing the pharmacological efficacy, stability, and controlled release of ginsenosides such as Rg3?

Reply: Advanced nanobiotechnological approaches in ginseng have significantly improved the bioactivity and pharmacokinetics of ginsenosides, as documented in multiple studies.

Li et al. (2010) showed improved cytotoxic and apoptotic activity of PEG–PLGA–Rg3 nanoparticles while better uptake, serum stability, and hybrid nano-complex NPs led to the longer circulation of (S)-Rg3 demonstrated in in vivo studies. Yang et al. (2014) developed an albumin-based nanocapsule loaded with ginsenoside Rg3, demonstrating higher efficiency. The combined effects of Rg3 chemotherapy and hyperthermia (nanosphere-induced magnetic fluid) were highly effective in the apoptosis of HeLa cervical cancer cells in vitro studies, suggesting that nanoencapsulation of ginsenoside Rg3 causes increases in stability, controlled release and potent efficacies of nanodrug.

Lines 248–252: What are the primary challenges associated with the cytotoxicity, solubility, and stability of ginsenoside formulations, and how can nanoconjugation strategies mitigate these issues?

Reply: While ginsenoside formulations are gaining recognition and marketed for their therapeutic properties, there are key limitations regarding the cytotoxicity, solubility and stability of ginseng formulations. While product substitution and adulteration have adversely affected the marketing of ginseng-derived products, the development of genetically pure and high-yielding varieties is necessary for the commercialization of ginseng products. In addition, the poor solubility and stability of ginsenosides (poor pharmacokinetics profile) is a major challenge in the further development of these drug-like molecules and it often fails in clinical trials.

Nanobiotechnological interventions in ginseng aim to address these key challenges in the development of ginseng formulations for treating human ailments. Nanoconjugation represents a method of choice to address these limitations-

Panax ginseng and its bioactive constituents (ginsenosides) are the key components for the development of nanoconjugates. In the study, ginsenoside Rb1/PPD-fabricated NPs were synthesized as the nano-ginseng delivery system, and it showed good biocompatibility and potent anticancer efficacy. In another example, PEG–PLGA–Rg3 nanoparticles (polymeric NPs) exhibited significant cytotoxicity and apoptosis in C6 glioma cells, stability, and better uptake in pharmacokinetic analysis.

Mathiyalagan and coworkers worked on ginsenoside CK and PPD (poor solubility) to improve the delivery by the hydrophilic-ginsenoside conjugates [107]. The conjugate was developed- ginsenoside C-3 position of the OH group was covalently linked to the amine group of hydrophilic glycol chitosan via linkage by carbodiimide coupling reactions. The study showed a rapid release of ginsenoside in acidic conditions (pH 5.0) like the tumor environment than at pH 7.4, in in vitro studies. In HT29 cancer cells, polymer ginsenoside conjugates demonstrated increased cytotoxicity and promoted cell viability than ginsenoside CK alone and hindered LPS-induced production of nitric oxide

Lines 380–385: What translational hurdles currently limit the clinical application of ginsenoside-based nanotherapeutics, and what future research directions could address these gaps?

Reply: Nanobiotechnology has witnessed considerable progress in healthcare with the development of advanced therapeutics for human disease treatment. However, currently, some translational hurdles hamper the progress of nanotherapeutics in clinical applications in terms of regulatory guidelines. For the clinical translation of nanotherapeutics, some criteria are to be taken into account in terms of early-stage product development, including quality features of the drug under consideration for safety and bioactivities, analytical methods used for product characterization, parameters of the process, and sufficient pre-clinical models for validation. Some other concerns include batch-to-batch consistency and scaling up the process needs to be monitored will facilitate a manufacturing process that is reproducible and in line with the regulatory guidelines.